# Advancing The Understanding Of Fixed Point Iterations In Loop Neural Networks: A Detailed Analytical Study

## Abstract

Recent empirical studies have identified fixed point iteration phenomena in deep neural networks, where the hidden state tends to stabilize after several layers, showing minimal change in subsequent layers. This observation has spurred the development of practical methodologies, such as accelerating inference by bypassing certain layers once the hidden state stabilizes, selectively fine-tuning layers to modify the iteration process, and implementing loops of specific layers to maintain fixed point iterations. Despite these advancements, the understanding of fixed point iterations remains superficial, particularly in high-dimensional spaces, due to the inadequacy of current analytical tools. In this study, we conduct a detailed analysis of fixed point iterations in a vector-valued function modeled by neural networks. We establish a sufficient condition for the existence of multiple fixed points in looped neural networks with varying input regions. Additionally, we expand our examination to include a robust version of fixed point iterations. To demonstrate the effectiveness and insights provided by our approach, we provide case studies that in looped neural networks, there may exist $2^d$ number of robust fixed points under exponentiation or polynomial activation functions, where $d$ is the feature dimension. Furthermore, our preliminary empirical results support our theoretical findings. Our methodology enriches the toolkit available for analyzing fixed-point iterations of loop neural networks and may enhance our comprehension of neural network mechanisms.

## 1 Introduction

Deep neural networks have achieved remarkable success and are widely employed in various applications, including ChatGPT (OpenAI, 2023), face recognition (Wang & Deng, 2021), and personalized recommendation systems (Da'u & Salim, 2020), among others. These networks typically consist of numerous hidden layers; for instance, Residual Networks (ResNets) (He et al., 2016) can contain over 1,000 layers. Recent empirical studies reveal that, despite the numerous layers in deep neural networks, certain operational phases exist where adjacent layers may perform identical operations (Meng et al., 2022; Shi et al., 2024). Consequently, we can focus on modifying specific associated layers during inference to enhance performance. Furthermore, the hidden states tend to stabilize after several adjacent layers, resulting in minimal changes in subsequent layers, allowing us to skip certain layers during inference (Elhoushi et al., 2024). Additional research indicates that a looped transformer—where its output is fed back into itself iteratively—exhibits expressive capabilities comparable to programmable computers (Giannou et al., 2023) and is more effective at learning algorithms (Yang et al., 2023). Other studies have also examined the convergence of deep neural networks when the weights across different layers are nearly identical, with only minor perturbations (Xu & Zhang, 2022; 2024). Collectively, these findings suggest the relevance of fixed-point iteration (Definition 3.1). Employing fixed-point methods in deep or looped neural networks may offer several advantages, such as reducing the number of parameters and dynamically adjusting runtime based on the complexity of the problem.

Despite these advancements, our understanding of fixed-point iterations remains limited, especially in high-dimensional spaces, due to the limitations of existing analytical tools. It remains challenging

to determine when a neural network can effectively approximate a fixed-point solution and how many layers or iterations are required to ensure a good outcome.

Thus, it is natural to ask the following question:

*How shall we analytically study fixed point iterations in deep neural networks?*

In this study, we conduct a detailed analysis of fixed point iterations in a vector-valued function, $\mathbb{R}^d \to \mathbb{R}^d$, where $d$ is the hidden feature size modeled by neural networks. We establish a general theorem (Theorem 4.1) to describe a sufficient condition for the existence of multiple fixed points of looped neural networks (Definition 3.5) based on varying input regions. Then, we expand our examination by introducing noise during each fixed point iteration and show that the fixed point iteration process is robust under noise (Theorem 4.2). It represents deep neural networks with residue connection (He et al., 2016), where after each layer, the hidden states are only perturbed slightly. Finally, we demonstrate the effectiveness of our approach by studying looped neural networks under polynomial and exponential activation functions (Theorem 5.1 and Theorem 5.2). We show that in the looped neural networks, there may exist $2^d$ number of robust fixed points. Recall that the previous tools can only handle single fixed point analysis (Joudaki & Hofmann, 2025), while our analysis can be applied to more practical cases. Furthermore, our preliminary empirical results support our theoretical findings (Section 6). Our methodology enriches the toolkit available for analyzing fixed point iterations of vector-valued functions and may help us better understand neural network mechanisms.

**Our contributions:**

- We study the fixed point iteration in looped neural networks and provide a general theorem (Theorem 4.1) to describe a sufficient condition for the existence of multiple fixed points. We also establish a robust version of fixed point iterations with noise perturbation (Theorem 4.2).
- We provide two case studies where looped neural networks may have $2^d$ number of robust fixed points, which demonstrates the effectiveness of our approach (Theorem 5.1 and Theorem 5.2). Our preliminary empirical results validate our theoretical findings (Section 6).

**Roadmap.** Our paper is organized as follows. In Section 2, we review related literature. In Section 3, we present the preliminary of our notations, Banach fixed point theorem, and our definition of a Looped Neural Network. In Section 4, we outline the main results of this work. In Section 5, we present case analysis results of fixed-point iterations for neural networks using two types of activation functions. In Section 6, we present the experimental results of this work. In Section 7, we conclude our paper.

## 2 RELATED WORK

In Section 2.1, we introduce fixed point theory with a focus on the Banach fixed point theorem. In Section 2.2, we present some work on incorporating looped structures into neural networks. In Section 2.3, we introduce some works that utilize the properties of fixed point iterations in neural network computations.

### 2.1 FIXED POINT ITERATION METHODS

In numerical analysis, fixed point iteration methods (Agarwal et al., 2001; Istratescu, 2001; Granas & Dugundji, 2003; Khojasteh et al., 2015) use the concept of fixed points to compute the solution of a given equation in a repetitive manner. Many works have focused on the convergence properties of fixed-point iteration methods. For example, the Banach fixed point theorem (Atkinson & Han, 2009) gives a sufficient condition under which a unique fixed point exists and is approachable via iterative methods. Although there are other fixed-point theorems, Banach fixed-point theorem, in particular, is useful because it provides a clear criterion for fixed points using contraction mappings. If a function is a contraction, the theorem guarantees the existence and uniqueness of a fixed point, making it easier to work with than other fixed-point theorems that may have more complex conditions.

Recent works focus on employing various methods to accelerate the convergence of fixed-point iterations. For instance, (Zhou et al., 2011) proposed a Quasi-Newton method for accelerating fixed-point iterations by approximating the Jacobian in Newton's method, enabling efficient root-finding for the function $g(x) = x - f(x)$. (Walker & Ni, 2011) presents Anderson acceleration, an under-utilized method for enhancing fixed-point iterations. After this work, (Zhang et al., 2020) introduces a globally convergent variant of type-I Anderson acceleration for non-smooth fixed-point problems, improving terminal convergence of first-order algorithms.

## 2.2 Looped Neural Networks

Looped Neural Networks are a paradigm in deep learning that aims to address certain limitations of traditional feedforward architectures. The addition of loopy structures to traditional neural networks has already received extensive research. For example, (Caswell et al., 2016) introduces a looped Convolutional Neural Network, which unrolls over multiple time steps and demonstrates that these networks outperform deep feedforward networks on some image datasets.

Transformers (Vaswani et al., 2017; Chu et al., 2023; Liang et al., 2024a) have become the preferred model of choice in natural language processing (NLP) and other domains that require sequence-to-sequence modeling. To understand why Transformers excel at iterative inference while lacking an iterative structure, (Giannou et al., 2023) proposes a Looped Transformer and has shown that transformer networks can be used as universal computers by programming them with specific weights and placing them in a loop. (Gatmiry et al., 2024) investigates the learnability of linear looped Transformers for linear regression, showing they can converge to algorithmic solutions via multi-step preconditioned gradient descent with adaptive preconditioners. (Gao et al., 2024; Fan et al., 2024; Xu & Sato, 2024; Chen et al., 2024; Liang et al., 2024b) argued that the Looped Transformer could achieve significantly higher algorithmic representation capabilities and good generalization ability while using the same number of parameters compared to the standard Transformer.

## 2.3 Neural Networks as Fixed Point Iterations

To better understand the convergence property and stability of neural networks, many works investigate neural networks as fixed-point iteration processes. The research in this area can be traced back to (Hyvärinen & Oja, 1997). This work demonstrates how a neural network learning rule can be converted into a fixed-point iteration, resulting in a simple, parameter-free algorithm that converges quickly to the optimal solution allowed by the data. Recently, many researchers have found that the hidden layers of many deep networks converge to a fixed point (a stable state). Based on this, treating neural networks as fixed points has received extensive research. For instance, (Yang et al., 2023) introduces a training method for looped transformers that emulates iterative algorithms, optimizing convergence with fewer parameters than standard transformers, which highlights that looped Transformers excel in learning tasks like in-context learning. (Joudaki & Hofmann, 2025) presents a framework to analyze kernel sequence evolution in neural networks, showing how hidden representations evolve and converge, with implications for activation functions and network design. (Bai et al., 2019) proposes the Deep Equilibrium Model (DEQ), which solves sequential data tasks by directly finding fixed points, bypassing iterative approximations.

## 3 Preliminary

We introduce some definitions that will be used throughout the paper in Section 3.1. Then, we briefly review the fixed point method and Banach fixed-point theorem in Section 3.2. Finally, we define the looped neural network in Section 3.3, which is the primary focus of our work.

### 3.1 Notations

For a vector $v \in \mathbb{R}^d$, we use $\|v\|_1, \|v\|_2$, and $\|v\|_\infty$ to denote the $\ell_1$-norm, $\ell_2$-norm, and $\ell_\infty$-norm of $v$, respectively. For two vectors $u, v \in \mathbb{R}^d$, we use $\langle u, v \rangle$ to denote the standard inner product of $u$ and $v$. We use $\mathbf{1}_d$ to denote a vector whose elements are all 1.

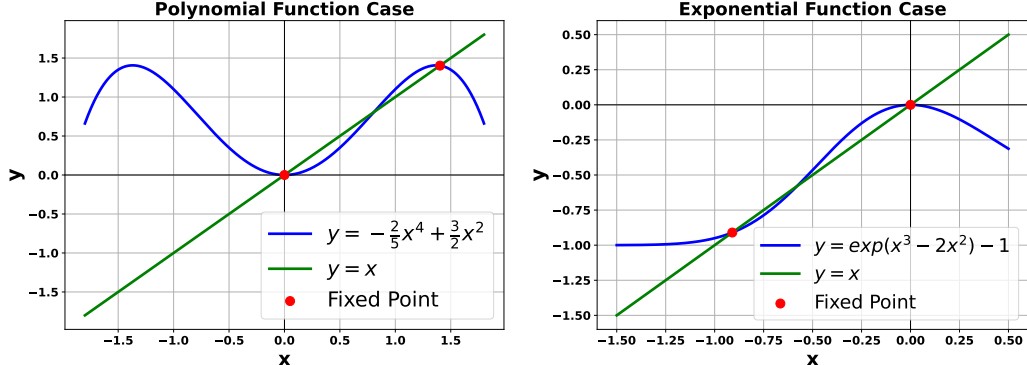

Figure 1: Example of polynomial (left) and exponential (right) functions contain at least two fixed points (red dots), and points near these fixed points will converge to them under fixed-point iteration.

## 3.2 Fixed Point Methods

In this section, we introduce the concept of the fixed-point method and the well-known Banach fixed-point theorem. Here, we only deal with the case where the space is $\mathbb{R}^d$. In Appendix A, we state the original definitions and theorems in the context of Banach spaces.

We first introduce the fixed point iteration problem.

**Definition 3.1** (Fixed point, (Atkinson & Han, 2009)). *Let $D$ be a subset of $\mathbb{R}^d$. We say a function $f : D \to \mathbb{R}^d$ has a fixed point $p \in D$ if $f(p) = p$.*

Then, we introduce contractive mapping, which is a key concept in fixed point iteration convergence.

**Definition 3.2** (Contractive mapping, Definition 5.1.2 of (Atkinson & Han, 2009)). *Let $\|\cdot\|$ be a norm on $\mathbb{R}^d$. Let $D$ be a subset of $\mathbb{R}^d$. We say that a function $f : D \to \mathbb{R}^d$ is contractive with contractivity constant $K \in [0, 1)$ if*

$$\|f(x) - f(x')\| \leq K\|x - x'\|, \ \forall x, x' \in V.$$

Contractive mapping means that the function's output space becomes 'smaller' than its input space. This allows us to introduce the Banach fixed-point theorem, the key tool in this work.

**Lemma 3.3** (Banach fixed-point theorem, Theorem 5.1.3 of (Atkinson & Han, 2009)). *Let $\|\cdot\|$ be a norm on $\mathbb{R}^d$. Let $D$ be a nonempty closed set of $\mathbb{R}^d$. Suppose that $f : D \to \mathbb{R}^d$ is a mapping that satisfies the following: (1) $f(x) \in D$ whenever $x \in D$. (2) $f$ is contractive with contractivity constant $K \in [0, 1)$.*

*Then, it holds that: (1) The function $f$ has a unique fixed point $p \in D$. (2) For any initial point $x^{(0)} \in D$, the fixed-point iteration $x^{(t)} = f(x^{(t-1)})$, $t \geq 1$, converges to the fixed point $p$ as $t \to \infty$. (3) The following error bounds hold:*

$$\|x^{(t)} - p\| \leq \min\{\frac{K^t}{1-K}\|x^{(1)} - x^{(0)}\|, \frac{K}{1-K}\|x^{(t)} - x^{(t-1)}\|, K\|x^{(t-1)} - p\|\}.$$

Lemma 3.3 told us the sufficient condition for a single unique fixed point. Although checking for contractivity is usually difficult, for differentiable functions, it becomes easier by examining the Jacobian matrix. The following lemma is a corollary of Lemma 3.3.

**Lemma 3.4** (Banach fixed point theorem, vector case, informal version of Lemma A.5). *Let $D \subseteq \mathbb{R}^d$ be a nonempty closed set. Suppose that $f : D \to \mathbb{R}^d$ is differentiable and satisfies the following: (1) $f(x) \in D$ whenever $x \in D$. (2) There exists constant $K < 1$ such that for every $i \in [d]$, $\|\frac{\mathrm{d}f_i(x)}{\mathrm{d}x}\|_1 \leq K$, $\forall x \in D$. where $f_i(x)$ is the $i$-th entry of $f(x)$. Then, it holds that (1) The function $f$ has a unique fixed point $p \in D$. (2) For any initial point $x^{(0)} \in D$, the fixed point iteration $x^{(t)} = f(x^{(t-1)})$, $t \geq 1$, converges to the fixed point $p$ as $t \to \infty$. (3) The below error bounds hold:*

$$\|x^{(t)} - p\|_\infty \leq \min\{\frac{K^t}{1-K}\|x^{(1)} - x^{(0)}\|_\infty, \frac{K}{1-K}\|x^{(t)} - x^{(t-1)}\|_\infty, K\|x^{(t-1)} - p\|_\infty\}.$$

When $d = 1$, Lemma 3.4 boils down to the scalar case.

### 3.3 LOOPED NEURAL NETWORKS

In this section, we give the formal definition of a Looped Neural Network.

**Definition 3.5** (Looped Neural Network). *Let $W = [w_1, \ldots, w_d]^\top \in \mathbb{R}^{d \times d}$ be a weight matrix and $b \in \mathbb{R}^d$ be the bias parameter. Let $g : \mathbb{R} \to \mathbb{R}$ be a differentiable activation function. We consider the one layer of neural network in the form*

$$f(x; W, b) := g(Wx + b)$$

*where $g$ is applied entry-wise. The L-layer looped neural network is defined as*

$$\mathsf{NN}(x^{(0)}; W, b, L) := x^{(L)}, \quad x^{(t)} := f(x^{(t-1)}; W, b), \ \forall t \in [L].$$

**Remark 3.6.** *Note that the Looped Neural Network (LNN) shares similarities with Recurrent Neural Networks (RNN) but is slightly different. RNN maps sequence to sequence, which means in each loop, RNN has new input data, while LNN does not. Furthermore, RNN generates output in each loop, while LNN only outputs after all loops.*

## 4 MAIN RESULTS

We first introduce our general theorem for looped neural networks in Section 4.1 and then introduce our robust version in Section 4.2.

### 4.1 GENERAL THEOREM

We have the following general theorem, which provides a sufficient condition for the existence of multiple fixed points for a Looped Neural Network.

**Theorem 4.1** (General result). *Consider the L-layer looped neural network $\mathsf{NN}(x^{(0)}; W, b, L)$ defined in Definition 3.5. If the following conditions hold: (1) There exists disjoint $D_1, \ldots, D_m \subseteq \mathbb{R}^d$ such that for every $i \in [m]$, and for every $x \in D_i$, $\mathsf{NN}(x; W, b, 1) \in D_i$. (2) The weight matrix $W$ and the activation function $g$ satisfy: For every $i \in [m]$, there exists $K_i \in [0, 1)$ such that for every $x \in D_i$, and for every $j \in [d]$, $|g'(\langle w_j, x \rangle)| \cdot \|w_j\|_1 \leq K_i$. Then, the following statements hold:*

- *The single layer of the looped neural network, $f(x; W)$, has at least $m$ fixed-points $p_1, \ldots, p_m$ satisfying : For every $i \in [m]$, there exists a constant $\epsilon_i > 0$, for any initial point $x^{(0)} \in D_i$ with $\|x^{(0)} - p_i\|_\infty \leq \epsilon_i$, we have*

$$\lim_{L \to \infty} \mathsf{NN}(x^{(0)}; W, b, L) = p_i.$$

- *For every $i \in [m]$, there exists a constant $c_i > 0$ such that for any $L \geq 2$,*

$$\|\mathsf{NN}(x^{(0)}; W, b, L) - p_i\|_\infty \leq K_i^L \cdot c_i \epsilon_i.$$

*Proof.* Assume the conditions in the statement hold. Fix $i \in [m]$. Then we have $f(x; W) = \mathsf{NN}(x; W, b, 1) \in D_i$ for every $x \in D_i$. Next, for every $j \in [d]$, and for the $j$-th entry of $f(x; W)$, where we denoted by $f_j(x; W)$, we have

$$\|\frac{\mathrm{d} f_j(x; W, b)}{\mathrm{d}x}\|_1 = \|\frac{dg(\langle w_j, x \rangle)}{dx}\|_1 = \|g'(\langle w_j, x \rangle) \cdot w_j\|_1 = |g'(\langle w_j, x \rangle)| \cdot \|w_j\|_1 \leq K_i,$$

where the first step follows from the definition of $f_j(x; W)$, the second step uses the chain rule, the third step is due to the property of norm, and the last step uses the condition in the lemma.

Therefore, we can apply Lemma 3.3, for each $i \in [m]$, there exists a fixed point $p_i$ which can be converged to by the fixed point iteration, and thus it can be found by the looped neural network with initial point $x^{(0)}$ when the number of layers goes to infinity. Next, for every $i \in [m]$, and for an initial point $x^{(0)} \in D_i$, there exists $\epsilon_i := \sup_{y, z \in D_i} \|y - z\|_\infty$, $c_i := \frac{1}{1 - K_i}$ such that we have

$$\|\mathsf{NN}(x^{(0)}; W, b, L) - p_i\|_\infty = \|x^{(L)} - p_i\|_\infty \leq \frac{K_i^t}{1 - K_i} \|x^{(1)} - x^{(0)}\|_\infty \leq K_i^L \cdot c_i \epsilon_i,$$

where the first step follows Definition 3.5, and the second step uses the error bound in Lemma 3.3, and the second steps follows the definition of $\epsilon_i$ and $c_i$. □

Theorem 4.1 gives us a way how to find different fixed points of vector-valued functions such as looped neural networks. Note that for different inputs, the fix point iteration behavior may be different, even when the model weights are fixed. We refer readers to Figure 1 and Figure 2 for more intuition.

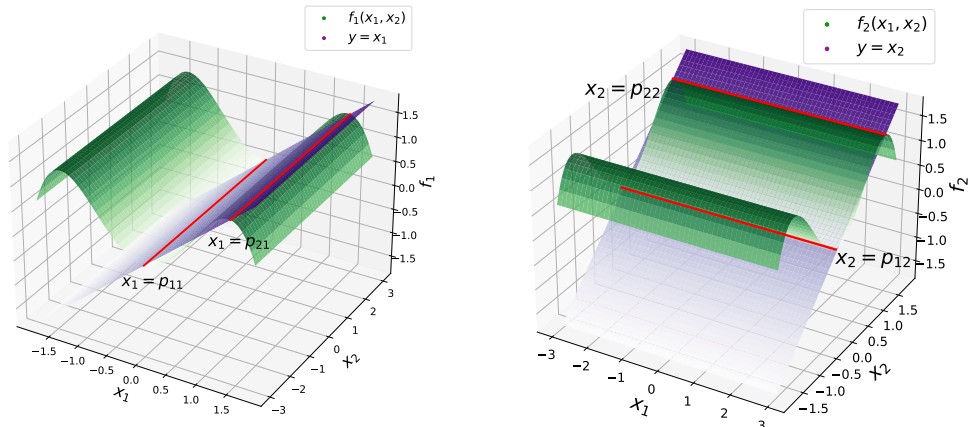

Figure 2: Example of a 2-$d$ case of Theorem 5.1. **Left**: The graph of $f_1(x_1, x_2)$ (in green) and $y = x_1$ (in purple), with the red line indicating the intersection of two curves (1-th dimension fixed point). **Right**: The graph of $f_2(x_1, x_2)$ (in green) and $y = x_2$ (in purple), with the red line indicating the intersection of two curves (2-th dimension fixed point).

### 4.2 PERTURBED FIXED POINT ITERATION

Next, we consider a variant of the fixed point method, where in each iteration, there is noise term $h(x)$.

**Theorem 4.2** (Robust Banach fixed point theorem, informal version of Lemma A.9). *Let $D \subseteq \mathbb{R}$ be a nonempty closed set. Suppose that $f : D \to \mathbb{R}$ is differentiable and satisfies the following: (1) $f(x) \in D$ whenever $x \in D$. (2) There exists constant $K \in [0, 0.95]$ such that $|f'(x)| \leq K$, $\forall x \in D$. (3) For any initial point $x^{(0)} \in D$, consider the perturbed fixed point iteration $x^{(t)} = f(x^{(t-1)}) + h(x^{(t-1)})$, for any $t \geq 1$, where the function $h$ satisfies $|h(x)| \leq 1/m$ for every $x \in D$ for some sufficiently large $m > 0$. Then, it holds that (1) The function $f$ has a unique fixed point $p \in D$. (2) The following error bounds hold:*

$$|x^{(t)} - p| \leq K|x^{(t-1)} - p| + \frac{1}{m} \ \ and \ \ |x^{(t)} - p| \leq K^t|x^{(0)} - p| + \frac{20}{m}.$$

Theorem 4.2 shows that each fix iteration process is robust and may not be hurt by the noise much. It corresponds to the setting that deep neural networks with residual connections (He et al., 2016), where after each layer, the hidden states are only slightly perturbed. Our experiments in Section 6 support our theoretical analysis of robustness.

## 5 CASE STUDY

In this section, we provide case studies of the Looped Neural Networks with different activation functions. We show that the Looped Neural Networks may have $2^d$ number of different fixed points.

First, we consider the polynomial activation function. We have the following robust fixed points statement.

**Theorem 5.1** (A specific polynomial activation function with small perturbation). *Consider the L-layer looped neural networks* $\mathsf{NN}(x^{(0)}; W, b, L)$ *defined in Definition 3.5. If the following conditions hold: (1) Let* $C := \sqrt{\frac{15 + \sqrt{65}}{8}}$. *(2) Let* $g(x) := -\frac{2}{5}x^4 + \frac{8}{5}Cx^3 + (\frac{3}{2} - \frac{12}{5}C^2)x^2 + (\frac{8}{5}C^3 - 3C)x + 1$ *denote the polynomial activation function used in this looped neural networks.*

*Then there exists a set of noisy parameters* $W$ *and* $b$ *such that* $x^{(t)} = \mathsf{NN}(x^{(t-1)}; W, b, L) + h(x^{(t-1)}) \in \mathbb{R}^d$, *for any* $t \in [L]$, *where the function* $h$ *satisfies* $|h(x)| \le 1/m$ *for every* $x \in \mathbb{R}^d$ *for some sufficiently large* $m > d$. *For this looped neural network, the following statements hold:*

- *The single layer of the looped neural network,* $f(x; W, b)$, *has at least* $2^d$ *robust fixed-points* $p_1, \cdots, p_{2^d}$ *satisfying: For every* $i \in [2^d]$, *there exists a vector* $\epsilon_i \in \mathbb{R}^d$, *for any initial point* $x^{(0)}$ *with* $\|x^{(0)} - p_i\|_\infty \le \|\epsilon_i\|_\infty$, *we have*

$$\lim_{L \to \infty} \mathsf{NN}(x^{(0)}; W, b, L) = p_i.$$

- *For every* $i \in [2^d]$, *there exists a constant* $c_i > 0$ *and a constant* $K_i \in [0, 0.9)$ *such that for any* $L \ge 2$, *we have*

$$\|\mathsf{NN}(x^{(0)}; W, b, L) - p_i\|_\infty \le K_i^t \cdot \|\epsilon_i\|_\infty + \frac{20}{m}.$$

*Proof.* Let $f(x; W, b)$ be a single layer version of $\mathsf{NN}(x; W, b, 1)$. Let $m > d$ be a sufficiently large constant. Let $W = \frac{1}{m^2}\mathbf{1}_d \cdot \mathbf{1}_d^\top + (-\frac{1}{m^2} + 1)\,\mathrm{diag}(\mathbf{1}_d) \in \mathbb{R}^{d \times d}$ denote the weight matrix. Let $b = [C, \cdots, C]^\top \in \mathbb{R}^d$ denote the bias vector. Then it's clear that for each $j \in [d]$, we have

$$g(\langle w_j, x \rangle + b_j) = -\frac{2}{5}x_j^4 + \frac{3}{2}x_j^2 + \frac{1}{m^2}h_j(x) = f_j(x; W, b) + \frac{1}{m^2}h_j(x)$$

where $h_j(x)$ is a polynomial in $x_1, \cdots, x_d$. For every $j \in [d]$, there exists a $D_1 = (1.302, 1.502)$ and $D_2 = (-0.3, 0.3)$, it satisfies that $f_j(x; W, b) \in D_1$ when $x_j \in D_1$ and $f_j(x; W, b) \in D_2$ when $x_j \in D_2$. By the proof in Lemma B.1, there exists a $K_j = 0.92$ such that for any $x_j \in D_1 \cup D_2$ we have $|f_j'(x; W, b)| \le K_j$.

It's obvious that when $m$ is sufficiently large and $\|\epsilon_i\|_\infty$ is small enough, we will have $|\frac{1}{m^2}h_j(x)| \le \frac{1}{m}$, where we can see $\frac{1}{m^2}h_j(x)$ as $h(x)$ in statement of Theorem 4.2. Then, by combing the result of Theorem 4.2, we have that for each dimension $j \in [d]$, we have 2 robust fixed points.

So we can get $f(x; W, b)$ has $2^d$ Robust Fixed Points trivially. The reason is that by our construction, for each dimension $j \in [d]$, the value of the function $f$ near the fixed point along the $j$-th dimension is only affected little by other dimensions. Hence, if each dimension admits two fixed points, then we have $2^d$ fixed points by considering all configurations.

And for any $i \in [2^d], j \in [d]$, we have

$$|\mathsf{NN}_j(x^{(t)}; W, b, L) - p_{i,j}| \le K_{i,j}^t |\mathsf{NN}_j(x^{(0)}; W, b, L) - p_{i,j}| + \frac{20}{m} \le K_{i,j}^t \epsilon_{i,j} + \frac{20}{m}$$

where the first step comes from the result of Theorem 4.2, and the second step comes from the range of values for the initial point. Then use the definition of $\|\cdot\|_\infty$, we have

$$\|\mathsf{NN}(x^{(t)}; W, b, L) - p_i\|_\infty \le K_i^t \cdot \|\epsilon_i\|_\infty + \frac{20}{m}.$$

Then, we complete the proof. $\qquad\square$

In Theorem 5.1, we can see that our Looped Neural Network has $2^d$ different robust fixed points solutions. Recall that the previous analysis tools can only handle single fixed point analysis. We can show the existence of $2^d$ different robust fixed points solutions when the Looped Neural Network uses exponential activation as well.

**Theorem 5.2** (A specific exponential activation function with small perturbation). *Consider the L-layer looped neural networks* $\mathsf{NN}(x^{(0)}; W, b, L)$ *defined in Definition 3.5. If the following conditions*

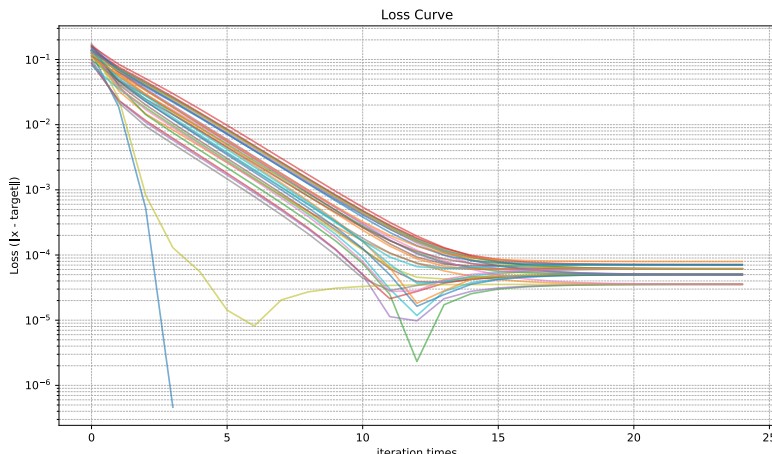

Figure 3: Empirical support for the Theorem 5.1 and Theorem 5.2. Let $d = 10$. We randomly pick $2^d = 1024$ points, where each point is in one of $2^d = 1024$ fixed points' neighborhoods with radius 1, i.e., $\|x - \text{target}\| \leq 1$. The $x$-axis is the number of looped iterations. The $y$-axis is the distance between the point at iteration $t$ and its corresponding fixed point. The results show that all points converge to their fixed points, and all $2^d = 1024$ fixed points are robust.

*hold: (1) Let $C := -2.15$. (2) Let $g(x) := \exp(x^3 + (-2 - 3C)x^2 + (3C^2 + 4C)x + \ln 2) - 1$ denote the exponential activation function used in this looped neural networks.*

*Then there exists a set of noisy parameters $W$ and $b$ such that $x^{(t)} = \mathsf{NN}(x^{(t-1)}; W, b, L) + h(x^{(t-1)}) \in \mathbb{R}^d$, for any $t \in [L]$, where the function $h$ satisfies $|h(x)| \leq 1/m$ for every $x \in \mathbb{R}^d$ for some sufficiently large $m > d$. For this looped neural network, the following statements hold:*

- *The single layer of the looped neural network, $f(x; W, b)$, has at least $2^d$ robust fixed-points $p_1, \cdots, p_{2^d}$ satisfying: For every $i \in [2^d]$, there exists a vector $\epsilon_i \in \mathbb{R}^d$, for any initial point $x^{(0)}$ with $\|x^{(0)} - p_i\|_\infty \leq \|\epsilon_i\|_\infty$, we have $\lim_{L \to \infty} \mathsf{NN}(x^{(0)}; W, b, L) = p_i$.*

- *For every $i \in [2^d]$, there exists a constant $c_i > 0$ and a constant $K_i \in [0, 0.9)$ such that for any $L \geq 2$, we have*

$$\|\mathsf{NN}(x^{(0)}; W, b, L) - p_i\|_\infty \leq K_i^t \cdot \|\epsilon_i\|_\infty + \frac{20}{m}.$$

*Proof.* Let $f(x; W, b)$ be a single layer version of $\mathsf{NN}(x; W, b, 1)$. Let $m > d$ be a sufficiently large constant. Let $W = \frac{1}{m^2} \mathbf{1}_d \cdot \mathbf{1}_d^\top + (-\frac{1}{m^2} + 1) \text{diag}(\mathbf{1}_d) \in \mathbb{R}^{d \times d}$ denote the weight matrix. Let $b = [C, \cdots, C]^\top \in \mathbb{R}^d$ denote the bias vector. Then it's clear that for each $j \in [d]$, we have

$$g(\langle w_j, x \rangle + b_j) = \exp(x_j^3 - 2x_j) - 1 + \frac{1}{m^2} h_j(x)$$

where the first step follows from $\exp(z) = 1 + O(z)$ when $|z| \leq 1$ and $|x_j| \leq 1$. $h_j(x)$ is an exponential function in $x_1, \cdots, x_d$. For every $j \in [d]$, there exists a $D_1 = (-0.1, 0.1)$ and $D_2 = (-1.01, -0.81)$, it satisfies that $f_j(x; W, b) \in D_1$ when $x_j \in D_1$ and $f_j(x; W, b) \in D_2$ when $x_j \in D_2$. By the proof in Lemma C.1, there exists a $K_j = 0.85$ such that for any $x_j \in D_1 \cup D_2$ we have $|f'_j(x; W, b)| \leq K_j$. Then, we finish the proof by following the same proof statement in Theorem 5.1. $\square$

Furthermore, our empirical results in Figure 3 support our Theorem 5.1 and Theorem 5.2, showing that there are $2^d$ fixed points in our looped neural networks.

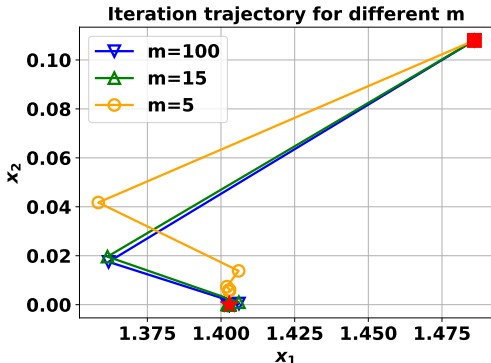

Figure 4: Different iteration trajectory when taking different $m$ in Theorem 4.2. We consider the 2-$d$ case, where the two axes are $x_1$ and $x_2$. The red square (■) in this figure is the initial point $x^{(0)} \in \mathbb{R}^2$. The red star (★) in this figure is the robust fixed point without noise. The $m$ represents the inverse of the noise level, i.e., noise $= 1/m$. From the figure, we can see that even though the noise changes the fixed point iteration process, different trajectories will eventually converge to the robust fixed point.

## 6 EXPERIMENTS

### 6.1 SETUP

In our experiments, we used Python 3.8 for simulation and employed Matplotlib 3.4.3 library to visualize the experimental results. We conduct three simulations in our work: (1) We are going to verify the polynomial function in Lemma B.1 and the exponential function in Lemma C.1 actually has 2 fixed point. (2) We are going to verify that the neural network in Theorem 5.1 has 2 fixed point in each dimension. (3) Recall that $m$ represents the inverse of the noise level in Theorem 5.1. We are going to verify that even the noise changes in the neural network of Theorem 5.1, different fixed point iteration trajectories will eventually converge to the Robust Fixed Point.

### 6.2 RESULTS

In Figure 1, we simulate the Banach fixed point iteration on a polynomial function $f(x) = -\frac{2}{5}x^4 + \frac{3}{2}x^2$ and an exponential function $f(x) = \exp(x^3 - 3x^2) - 1$. It shows that there are two fixed points ($x_1 = 0$ and $x_2 \approx 1.403$) of the polynomial function, and the points in the neighborhood of a fixed point will iterate to the corresponding fixed point through fixed-point iteration. It also shows that there are two fixed points ($x_1 = 0$ and $x_2 \approx -0.910$) of the exponential function, and the points in the neighborhood of a fixed point will iterate to the corresponding fixed point through fixed-point iteration. In Figure 2, we simulate a 2-$d$ neural network in Theorem 5.1. It shows that in the first dimension of the neural network, we have two robust fixed points ($x_1 = 0$ and $x_1 \approx 1.403$), and in the second dimension of the neural network, we have two robust fixed points ($x_2 = 0$ and $x_2 \approx 1.403$). In Figure 4, we simulate the effect of different values of $m$ on the fixed-point iteration. It shows that even though the noise changes the fixed point iteration process, different trajectories, $m = 5$, $m = 15$, and $m = 100$, when taking the same initial point (■), will eventually converge to the robust fixed point (★).

## 7 CONCLUSION

We provide an analytical framework for understanding fixed-point iterations in loop neural networks. We established theorems for multiple fixed points and their robustness under noise. Case studies with polynomial and exponential activation functions show the looped neural networks may have an exponential number of robust fixed points, demonstrating our approach's effectiveness. Our findings offer new insights into loop neural networks, contributing to the analysis toolkit and potentially enhancing loop neural network understanding and efficient algorithm design.

## ETHIC STATEMENT

This paper does not involve human subjects, personally identifiable data, or sensitive applications. We do not foresee direct ethical risks. We follow the ICLR Code of Ethics and affirm that all aspects of this research comply with the principles of fairness, transparency, and integrity.

## REPRODUCIBILITY STATEMENT

We ensure reproducibility on both theoretical and empirical fronts. For theory, we include all formal assumptions, definitions, and complete proofs in the appendix. For experiments, we describe model architectures and training details in the main text and appendix. Code and scripts are provided in the supplementary materials to replicate the empirical results.

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

# Appendix

**Roadmap.** In Section A, we introduce the fixed point method and the well-known Banach fixed-point theorem. In Section B, we provide the analysis of fixed points under the polynomial activation function. In Section C, we provide the analysis of fixed points under the exponential activation function.

## A   TOOLS OF FIXED POINT METHODS

In this section, we present the tools of fixed point methods used in our work. In Section A.1, we introduce the general case of Banach fixed point theorem. In Section A.2, we present the scalar case of Banach fixed point theorem. In Section A.3, we present the vector case of Banach fixed point theorem. In Section A.4, we present the matrix case of Banach fixed point theorem. In Section A.5, we present some calculation examples using Banach fixed point theorem.

### A.1   BANACH FIXED POINT THEOREM

We firstly present the definition of fixed point.

**Definition A.1** (Fixed point, (Atkinson & Han, 2009)). *Let $V$ be a Banach space with the norm $\| \cdot \|_V$, and let $D$ be a subset of $V$. We say a function $f : D \to V$ has a fixed point $p \in D$ if $f(p) = p$.*

And we introduce the concept of contractive mapping which will be used later.

**Definition A.2** (Contractive mapping, Definition 5.1.2 of (Atkinson & Han, 2009)). *Let $V$ be a Banach space with the norm $\| \cdot \|_V$, and let $D$ be a subset of $V$. We say that a function $f : D \to V$ is contractive with contractivity constant $K \in [0, 1)$ if*

$$\|f(x) - f(x')\|_V \le K\|x - x'\|_V, \ \forall x, x' \in V.$$

Now, we can present the Banach fixed point theorem, which is crucial in guaranteeing the existence and uniqueness of fixed points in metric spaces under contractive mappings, forming the foundation of many applications in analysis and applied mathematics.

**Lemma A.3** (Banach fixed point theorem, Theorem 5.1.3 of (Atkinson & Han, 2009)). *Let $V$ be a Banach space $V$ with the norm $\| \cdot \|_V$. Let $D$ be a nonempty closed set of $V$. Suppose that $f : D \to V$ is a mapping that satisfies the following*

- *$f(x) \in D$ whenever $x \in D$.*

- *$f$ is contractive with contractivity constant $K \in [0, 1)$.*

*Then it holds that*

- *The function $f$ has a unique fixed point $p \in D$.*

- *For any initial point $x^{(0)} \in D$, the fixed-point iteration $x^{(t)} = f(x^{(t-1)})$, $t \ge 1$, converges to the fixed point $p$ as $t \to \infty$.*

- *The following error bounds hold:*

$$\|x^{(t)} - p\|_V \le \frac{K^t}{1 - K}\|x^{(1)} - x^{(0)}\|_V,$$

$$\|x^{(t)} - p\|_V \le \frac{K}{1 - K}\|x^{(t)} - x^{(t-1)}\|_V,$$

$$\|x^{(t)} - p\|_V \le K\|x^{(t-1)} - p\|_V.$$

### A.2   SCALAR CASE

In this section, we present the scalar case of Banach fixed point theorem.

**Lemma A.4** (Banach fixed point theorem, scalar case). *Let $D \subseteq \mathbb{R}$ be a nonempty closed set. Suppose that $f : D \to \mathbb{R}$ is differentiable and satisfies the following:*

- *$f(x) \in D$ whenever $x \in D$.*

- *There exists constant $K < 1$ such that*

$$|f'(x)| \leq K, \ \forall x \in D.$$

*Then it holds that*

- *The function $f$ has a unique fixed point $p \in D$.*

- *For any initial point $x^{(0)} \in D$, the fixed point iteration $x^{(t)} = f(x^{(t-1)})$, $t \geq 1$, converges to the fixed point $p$ as $t \to \infty$.*

- *The following error bounds hold:*

$$|x^{(t)} - p| \leq \frac{K^t}{1-K}|x^{(1)} - x^{(0)}|,$$

$$|x^{(t)} - p| \leq \frac{K}{1-K}|x^{(t)} - x^{(t-1)}|,$$

$$|x^{(t)} - p| \leq K|x^{(t-1)} - p|.$$

*Proof.* It directly holds by Lemma A.5 with $d = 1$. $\square$

### A.3 VECTOR CASE

In this section, we extend the Banach fixed point theorem to its vector version. Note that $\mathbb{R}^d$ is a Banach space with norm $\|\cdot\|_\infty$. The $\ell_\infty$ norm of $x \in \mathbb{R}^d$ is defined as $\|x\|_\infty = \max_{i \in [d]} |x_i|$.

**Lemma A.5** (Banach fixed point theorem, vector case, formal version of Lemma 3.4). *Let $D \subseteq \mathbb{R}^d$ be a nonempty closed set. Suppose that $f : D \to \mathbb{R}^d$ is differentiable and satisfies the following:*

- *$f(x) \in D$ whenever $x \in D$.*

- *There exists constant $K < 1$ such that for every $i \in [d]$,*

$$\|\frac{\mathrm{d}f_i(x)}{\mathrm{d}x}\|_1 \leq K \ \forall x \in D.$$

*Then it holds that*

- *The function $f$ has a unique fixed point $p \in D$.*

- *For any initial point $x^{(0)} \in D$, the fixed point iteration $x^{(t)} = f(x^{(t-1)})$, $t \geq 1$, converges to the fixed point $p$ as $t \to \infty$.*

- *The following error bounds hold:*

$$\|x^{(t)} - p\|_\infty \leq \frac{K^t}{1-K}\|x^{(1)} - x^{(0)}\|_\infty,$$

$$\|x^{(t)} - p\|_\infty \leq \frac{K}{1-K}\|x^{(t)} - x^{(t-1)}\|_\infty,$$

$$\|x^{(t)} - p\|_\infty \leq K\|x^{(t-1)} - p\|_\infty.$$

*Proof.* We only need to show that $f$ is a contraction. For any $y, y' \in \mathbb{R}^d$, for any $i \in [d]$, there exists $z$ on the line segament between $y$ and $y'$ such that

$$|f_i(y) - f_i(y')| \leq |\langle \frac{\mathrm{d}f_i(x)}{\mathrm{d}x}|_{x=z}, y - y'\rangle|$$

$$\leq \|\frac{\mathrm{d}f_i(x)}{\mathrm{d}x}|_{x=z}\|_1, \|y - y'\|_\infty$$

$$\leq K\|y - y'\|_\infty.$$

where the first step follows from mean value theorem, the second step uses the Holder's inequality, and the last step is due to the second condition.

Therefore, by definition of $\|\cdot\|_\infty$, we have

$$\|f(y) - f(y')\|_\infty \leq K\|y - y'\|_\infty.$$

Hence $f$ is contractive with conctractivity constant $K \in [0, 1)$. By Lemma A.3, the proof is complete. $\square$

### A.4 MATRIX CASE

In this section, we extend the Banach fixed point theorem to its matrix version. Note that $\mathbb{R}^{n \times d}$ is a Banach space with norm $\|\cdot\|_\infty$. The maximum norm of $A \in \mathbb{R}^{n \times d}$ is defined as $\|A\|_\infty = \max_{i \in [n], j \in [d]} |A_{i,j}|$.

**Lemma A.6** (Banach fixed point theorem). *Let $D \subseteq \mathbb{R}^{n \times d}$ be a nonempty closed set. Suppose that $f : D \to \mathbb{R}^{n \times d}$ is differentiable and satisfies the following:*

- *$f(X) \in D$ whenever $X \in D$.*

- *There exists constant $K < 1$ such that for every $i, k \in [n]$ and $j, l \in [d]$*

$$|\frac{\mathrm{d}f(X)_{i,j}}{\mathrm{d}X_{k,l}}| \leq \frac{K}{nd}, \ \forall X \in D.$$

*Then it holds that*

- *The function $f$ has a unique fixed point $p \in D$.*

- *For any initial point $x^{(0)} \in D$, the fixed point iteration $x^{(t)} = f(x^{(t-1)})$, $t \geq 1$, converges to the fixed point $p$ as $t \to \infty$.*

- *The following error bounds hold:*

$$|x^{(t)} - p| \leq \frac{K^t}{1 - K}|x^{(1)} - x^{(0)}|,$$

$$|x^{(t)} - p| \leq \frac{K}{1 - K}|x^{(t)} - x^{(t-1)}|,$$

$$|x^{(t)} - p| \leq K|x^{(t-1)} - p|.$$

*Proof.* Here we only need to show that $f$ is a contraction. For any $X, X' \in \mathbb{R}^{n \times d}$, for any $i \in [n], j \in [d]$, there is a $Z$ between the matrix $X$ and $X'$ such that

$$|f(X)_{i,j} - f(X')_{i,j}| \leq |\frac{\mathrm{d}f(X)_{i,j}}{\mathrm{d}X}|_{X=Z} \circ (X - X')|$$

$$\leq |\frac{K}{nd}(\mathbf{1}_{n \times d} \circ (X - X'))|$$

$$\leq K\|X - X'\|_\infty$$

where the first step follows from mean value theorem, the second step uses the second condition in the lemma, and the last step is due to basic algebra.

Therefore, by definition of $\|\cdot\|_\infty$, we have

$$\|f(X) - f(X')\|_\infty \leq K\|X - X'\|_\infty$$

Hence $f$ is contractive with contractivity constant $K \in [0, 1)$. By Lemma A.3, the proof is complete.

$\square$

## A.5 Examples Using Fixed Point Theorem

In this section, we present some calculation case by using Banach fixed point theorem.

**Theorem A.7.** *Fixed Point Theorem:*

- *If $g \in C[a, b]$ and $a \le g(x) \le b$ for all $x \in [a, b]$, then $g$ has least one fixed point in $[a, b]$*

- *If, in addition, $g'$ exists in $[a, b]$, and $\exists k < 1$ such that $|g'(x)| \le k < 1$ for all $x$, then $g$ has a unique fixed point in $[a, b]$.*

**Success Case**: $g(x) = \frac{x^2 - 1}{3}$ has a unique fixed point in $[-1, 1]$.

*Proof.* First we need show $g(x) \in [-1, 1]$, $\forall x \in [-1, 1]$. Find the max and min values of $g$ as $-\frac{1}{3}$ and 0. (Hint: find critical points of $g$ first). So $g(x) \in [-\frac{1}{3}, 0] \subset [-1, 1]$.

Also $|g'(x) = \frac{2x}{3}| \le \frac{2}{3} < 1$, $\forall x \in [-1, 1]$, so $g$ has unique fixed point in $[-1, 1]$ by Fixed Point Theorem. $\square$

**Failed Case 1**: $g(x) = \frac{x^2 - 1}{3}$ has a unique fixed point in $[3, 4]$. But we can't use FPT to show this.

Note that there is a unique fixed point in $[3, 4](p = \frac{3 + \sqrt{13}}{2})$, but $g(4) = 5 \notin [3, 4]$, and $g'(4) = \frac{8}{3} > 1$ so we cannot apply FPT here.

From this example, we know FPT provides a sufficient but not necessary condition.

**Failed Case 2**: We can use FPT to show that $g(x) = 3^{-x}$ must have Fixed Point on $[0, 1]$, but we can't use FPT to show if it's unique (even though the FP on $[0, 1]$ is unique in this example).

Solution. $g'(x) = (3^{-x})' = -3^{-x} \ln 3 < 0$, therefore $g(x)$ is strictly decreasing on $[0, 1]$. Also $g(0) = 3^0 = 1$ and $g(1) = 3^{-1}$, so $g(x) \in [0, 1]$, $\forall x \in [0, 1]$. So a FP exists by FPT.

However, $g'(0) = -\ln 3 \approx -1.098$, so we don't have $|g'(x)| < 1$ over $[0, 1]$. Hence FPT does not apply.

Nevertheless, the FP must be unique since $g$ strictly decreases and intercepts with $y = x$ line only once.

Then We will present a lemma that proves a quadratic polynomial function cannot have two fixed points that satisfy the conditions of the Banach fixed point theorem.

**Lemma A.8.** *There doesn't exist a quadratic function that contains two fixed points and simultaneously satisfies that points in the neighborhood of the fixed points can converge to the respective fixed point through iteration.*

*Proof.* For $\forall a, b, c \in R$, let $f(x) = ax^2 + bx + c$ be a quadratic function. Then we have $f'(x) = 2ax + b$. Assuming that $f(x)$ has two fixed points, then we have:

$$x_1 = \frac{(1 - b) + \sqrt{(b - 1)^2 - 4ac}}{2a}, x_2 = \frac{(1 - b) - \sqrt{(b - 1)^2 - 4ac}}{2a}$$

Then we have,

$$f'(x_1) = 1 + \sqrt{(b - 1)^2 - 4ac} > 1$$

$$f'(x_2) = 1 - \sqrt{(b - 1)^2 - 4ac} < 1$$

So there is only one fixed point can satisfy the second condition of Lemma A.3. $\square$

## A.6 Robust Banach fixed point theory

**Theorem A.9** (Robust Banach fixed point theorem, scalar case, formal version of Lemma 4.2)**.** *Let $D \subseteq \mathbb{R}$ be a nonempty closed set. Suppose that $f : D \to \mathbb{R}$ is differentiable and satisfies the following: (1) $f(x) \in D$ whenever $x \in D$. (2) There exists constant $K \in [0, 0.95]$ such that*

$|f'(x)| \leq K$, $\forall x \in D$. *(3) For any initial point $x^{(0)} \in D$, consider the perturbed fixed point iteration $x^{(t)} = f(x^{(t-1)}) + h(x^{(t-1)})$, for any $t \geq 1$, where the function $h$ satisfies $|h(x)| \leq 1/m$ for every $x \in D$ for some sufficiently large $m > 0$.*

*Then, it holds that (1) The function $f$ has a unique fixed point $p \in D$. (2) The following error bounds hold:*

$$|x^{(t)} - p| \leq K|x^{(t-1)} - p| + \frac{1}{m} \text{ and } |x^{(t)} - p| \leq K^t|x^{(0)} - p| + \frac{20}{m}.$$

*Proof.* Clearly $f$ has a fixed point $p \in D$ by Lemma A.4.

We first show the first error bound. We can show that

$$\begin{aligned}
|x^{(t)} - p| &= |f(x^{(t-1)}) - p + h(x^{(t-1)})| \\
&= |f(x^{(t-1)}) - f(p) + h(x^{(t-1)})| \\
&= |f'(c) \cdot (x^{(t-1)} - p) + h(x^{(t-1)})| \\
&\leq |f'(c) \cdot (x^{(t-1)} - p)| + |h(x^{(t-1)})| \\
&= |f'(c)| \cdot |x^{(t-1)} - p| + |h(x^{(t-1)})| \\
&\leq K|x^{(t-1)} - p| + \frac{1}{m}
\end{aligned}$$

where the first step uses the perturbed fixed point iteration $x^{(t)} = f(x^{(t-1)}) + h(x^{(t-1)})$, the second step is due to the fact that $p$ is a fixed point of $f$, the third step follows from the mean value theorem where $c$ is a point between $x^{(t-1)}$ and $p$, the fourth step uses the triangle inequality, the fifth step follows from basic algebra, and the last step follows from $|f'(x)| \leq K$ and $|h(x)| \leq 1/m$.

Next, we show the second error bound. We can show that

$$\begin{aligned}
|x^{(t)} - p| &\leq K|x^{(t-1)} - p| + \frac{1}{m} \\
&\leq K(K|x^{(t-2)} - p| + \frac{1}{m}) + \frac{1}{m} \\
&\leq \cdots \\
&\leq K^t|x^{(0)} - p| + \sum_{i=1}^{t-1} \frac{K^i}{m} \\
&\leq K^t|x^{(0)} - p| + \frac{1 - K^{t-1}}{(1 - K)m} \\
&\leq K^t|x^{(0)} - p| + \frac{20}{m},
\end{aligned}$$

where the first four steps follow from recursively using the third error bound, the fifth step is due to the sum of geometric series, and the last step follows from $K \in [0, 0.95]$. $\qquad\square$

## B  CASE STUDY: POLYNOMIAL ACTIVATION

In this section, we start with a simple example of a single variable function which has at least two fixed points that can be found using the fixed point method. Then we use it to construct a specific neural network with polynomial activation function. Finally, we show that this neural network has at least two fixed points that can be found using the fixed point method.

### B.1  POLYNOMIAL ACTIVATION

Firstly, we present a univariate polynomial function that has at least two fixed points, which can be identified using the fixed point method.

**Lemma B.1.** *Let $f : \mathbb{R} \to \mathbb{R}$ be a function defined as $f(x) := -\frac{2}{5}x^4 + \frac{3}{2}x^2$. Then the following statements hold:*

- *The function $f$ has at least two fixed points which can be found by the fixed-point iteration, and we denoted them as $p_1, p_2$.*

- *For $i \in \{1, 2\}$, there exists a constant $\epsilon_i > 0$, for any initial point $x^{(0)} \in [p_i - \epsilon_i, p_i + \epsilon_i]$, the fixed-point iteration $x^{(t)} = f(x^{(t-1)})$ converges to the fixed point $p_i$.*

- *For $i \in \{1, 2\}$, there exists a constant $c_i > 0$ and a constant $K_i \in [0, 1)$ such that for any $t \geq 2$,*

$$|x^{(t)} - p_i| \leq K_i^t \cdot c_i \epsilon_i.$$

*Proof.* Clearly, $p_1 = 0$ and $p_2 \approx 1.4028$ are two fixed points of $f$. We show that $p_1$ and $p_2$ can be found by the fixed-point iteration and the error bounds hold.

For the fixed point $p_1 = 0$, let $\epsilon_1 = 0.3$, $K_1 = 0.9$, and $C_1 = 2/(1 - K_1)$. For any $x \in [p_1 - \epsilon_1, p_1 + \epsilon_1] = [-0.3, 0.3]$, we have $f(x) \in [0, 0.132] \subseteq [-0.3, 0.3]$. Hence the first condition of Lemma A.4 is satisfied. For any $x \in [p_1 - \epsilon_1, p_1 + \epsilon_1]$ we have $|f'(x)| < K_1 = 0.9 < 1$. Hence the second condition of Lemma A.4 is satisfied. Thus by Lemma A.4, if we pick the initial point $x^{(0)} \in [p_1 - \epsilon_1, p_1 + \epsilon_1]$, we can find $p_1$ by the fixed point iteration $x^{(t)} = f(x^{(t-1)})$, and it holds that for any $t \geq 2$,

$$
\begin{aligned}
|x^{(t)} - p_1| &\leq \frac{K_1^t}{1 - K_1} |x^{(1)} - x^{(0)}| \\
&\leq \frac{K_1^t}{1 - K_1} \cdot 2\epsilon_1 \\
&= K_1^t \cdot C_1 \epsilon_1,
\end{aligned}
$$

where the first step follows from Lemma A.4, the second step follows from $x^{(0)}, x^{(1)} \in [p_1 - \epsilon_1, p_1 + \epsilon_1]$, and the last step follows from $c_1 = 2/(1 - K_1)$.

For the fixed point $x_2 \approx 1.4028$, let $\epsilon_2 = 0.1$, $K_2 = 0.92$, and $C_2 = 2/(1 - K_2)$. For any $x \in [p_2 - \epsilon_2, p_2 + \epsilon_2] = [1.3028, 1.5028]$, we have $f(x) \in [1.347, 1.397] \subseteq [1.3028, 1.5028]$. Hence the first condition of Lemma A.4 is satisfied. For any $x \in [p_2 - \epsilon_2, p_2 + \epsilon_2]$, we have $|f'(x)| < K_2 = 0.92 < 1$. Hence the second condition of Lemma A.4 is satisfied. Thus by Lemma A.4, if we pick the initial point $x^{(0)} \in [p_2 - \epsilon_2, p_2 + \epsilon_2]$, we can find $p_2$ by the fixed point iteration $x^{(t)} = f(x^{(t-1)})$, and it holds that for any $t \geq 2$,

$$
\begin{aligned}
|x^{(t)} - p_2| &\leq \frac{K_2^t}{1 - K_2} |x^{(1)} - x^{(0)}| \\
&\leq \frac{K_2^t}{1 - K_2} \cdot 2\epsilon_2 \\
&= K_2^t \cdot c_2 \epsilon_2,
\end{aligned}
$$

where the first step follows from Lemma A.4, the second step follows from $x^{(0)}, x^{(1)} \in [p_2 - \epsilon_2, p_2 + \epsilon_2]$, and the last step follows from $c_2 = 2/(1 - K_2)$.

Therefore, we complete the proof. $\qquad \square$

Next, we prove two lemmas which are useful to construct a neural network with a polynomial acitivation function.

**Lemma B.2.** *Let*

$$f_1(x) := -\frac{2}{5}\langle u, x \rangle^4 + \frac{8}{5}C\langle u, x \rangle^3 + (\frac{3}{2} - \frac{12}{5}C^2)\langle u, x \rangle^2 + (\frac{8}{5}C^3 - 3C)\langle u, x \rangle + 1,$$

*where $C := \sqrt{\frac{15 + \sqrt{65}}{8}}$, $u := [1, 1, C - 1]$. Let $f_2(x) := 1, f_3(x) := 1$. Then there exists a polynomial $g(z) := a_4 z^4 + a_3 z^3 + a_2 z^2 + a_1 z + a_0$ where $a_0, a_1, a_2, a_3, a_4 \in \mathbb{R}$ such that for some $w_1, w_2, w_3 \in \mathbb{R}^3$, we have*

$$g(\langle w_1, x \rangle) = f_1(x),$$

$$g(\langle w_2, x\rangle) = f_2(x),$$
$$g(\langle w_3, x\rangle) = f_3(x).$$

*Moreover we have*

$$g(z) = -\frac{2}{5}z^4 + \frac{8}{5}Cz^3 + (\frac{3}{2} - \frac{12}{5}C^2)z^2 + (\frac{8}{5}C^3 - 3C)z + 1,$$

*and* $w_1 = [1, 1, C-1]^\top, w_2 = [0, 0, 0]^\top, w_3 = [0, 0, 0]^\top.$

*Proof.* Let $g(z) := -\frac{2}{5}z^4 + \frac{8}{5}Cz^3 + (\frac{3}{2} - \frac{12}{5}C^2)z^2 + (\frac{8}{5}C^3 - 3C)z + 1.$ Let $w_1 := [1, 1, 1 - C]^\top, w_2 := [0, 0, 0]^\top, w_3 := [0, 0, 0]^\top.$ Then it is clear that we have

$$g(\langle w_1, x\rangle) = f_1(x),$$
$$g(\langle w_2, x\rangle) = f_2(x),$$
$$g(\langle w_3, x\rangle) = f_3(x).$$

Thus we complete the proof. $\qquad\square$

**Lemma B.3.** *Let*

$$f_1(x) := -\frac{2}{5}\langle u, x\rangle^4 + \frac{8}{5}C\langle u, x\rangle^3 + (\frac{3}{2} - \frac{12}{5}C^2)\langle u, x\rangle^2 + (\frac{8}{5}C^3 - 3C)\langle u, x\rangle + 1,$$

*where* $C := \sqrt{\frac{15 + \sqrt{65}}{8}}, u := [1, 1, C-1].$ *It holds that*

$$f_1([x_1, 1, 1]^\top) = -\frac{2}{5}x_1^4 + \frac{3}{2}x_1^2.$$

*Proof.* Note that $\langle u, [x_1, 1, 1]^\top\rangle = x_1 + C.$ Hence

$$f_1([x_1, 1, 1]^\top)$$
$$= -\frac{2}{5}(x_1 + C)^4 + \frac{8}{5}C(x_1 + C)^3 + (\frac{3}{2} - \frac{12}{5}C^2)(x_1 + C)^2 + (\frac{8}{5}C^3 - 3C)(x_1 + C) + 1$$
$$= -\frac{2}{5}x_1^4 + \frac{3}{2}x_1^2,$$

where the first step follows from $\langle u, [x_1, 1, 1]^\top\rangle = x_1 + C$, and the second step follows from expanding the higher-order terms and simplifying it. $\qquad\square$

## B.2 EXTEND TO NEURAL NETWORKS

We are now prepared to construct a neural network with polynomial activation functions. By carefully designing the weight matrix, we can ensure that the network converges to different fixed points for different initilization.

**Lemma B.4.** *Let* $W := [w_1, w_2, w_3] \in \mathbb{R}^{3\times 3}$ *be a weight matrix. Let* $C = \sqrt{\frac{15 + \sqrt{65}}{8}}$ *be a constant. Let a polynomial activation function* $g : \mathbb{R} \to \mathbb{R}$ *be defined as*

$$g(z) = -\frac{2}{5}z^4 + \frac{8}{5}Cz^3 + (\frac{3}{2} - \frac{12}{5}C^2)z^2 + (\frac{8}{5}C^3 - 3C)z + 1,$$

*We define one layer of a neural network, denoted as* $f$, *as follows:*

$$f(x; W) := [g(\langle w_1, x\rangle), g(\langle w_2, x\rangle), g(\langle w_3, x\rangle)]^\top.$$

*Then there exists a weight matrix* $W$ *following statements hold:*

- *The function* $f(x; W)$ *has at least two fixed points which can be found by the fixed-point iteration, and we denoted them as* $p_1, p_2$.

- *For* $i \in \{1, 2\}$, *there exists a constant* $\epsilon_i > 0$, *for any initial point* $x^{(0)} \in [p_{i,1} - \epsilon_i, p_{i,1} + \epsilon_i] \times \{1\} \times \{1\}$, *the fixed-point iteration* $x^{(t)} = f(x^{(t-1)}; W)$ *converges to the fixed point* $p_i$.

- *For $i \in \{1, 2\}$, there exists a constant $c_i > 0$ and a constant $K_i \in [0, 1)$ such that for any $t \geq 2$,*

$$\|x^{(t)} - p_i\|_\infty \leq K_i^t \cdot c_i \epsilon_i.$$

*Proof.* The lemma holds by combining Lemma B.1, Lemma B.2, and Lemma B.3. $\square$

Then we can further extend the Lemma B.4 to the version without dummy variables by carefully designing the weight matrix.

**Lemma B.5** (A specific setting without dummy variables)**.** *Let $W := [w_1, w_2, w_3] \in \mathbb{R}^{3 \times 3}$ be a weight matrix. Let $b := [b_1, b_2, b_3] \in \mathbb{R}^3$ be a bias matrix. Let $C = \sqrt{\frac{15 + \sqrt{65}}{8}}$ be a constant. Let a polynomial activation function $g : \mathbb{R} \to \mathbb{R}$ be defined as*

$$g(z) := -\frac{2}{5}z^4 + \frac{8}{5}Cz^3 + (\frac{3}{2} - \frac{12}{5}C^2)z^2 + (\frac{8}{5}C^3 - 3C)z + 1.$$

*We define one layer of a neural network, denoted as $f$, as follows:*

$$f(x; W) := [g(\langle w_1, x \rangle + b_1), g(\langle w_2, x \rangle + b_2), g(\langle w_3, x \rangle + b_3)]^\top.$$

*Then there exists a weight matrix $W$ and a bias matrix $b$ following statements hold:*

- *The function $f(x; W)$ has at least two fixed points which can be found by the fixed-point iteration, and we denoted them as $p_1, p_2$.*

- *For $i \in \{1, 2\}$, there exists a constant $\epsilon_i > 0$, for any initial point $x^{(0)} \in [p_{i,1} - \epsilon_{i,1}, p_{i,1} + \epsilon_{i,1}] \times [p_{i,2} - \epsilon_{i,2}, p_{i,2} + \epsilon_{i,2}] \times [p_{i,3} - \epsilon_{i,3}, p_{i,3} + \epsilon_{i,3}]$, the fixed-point iteration $x^{(t)} = f(x^{(t-1)}; W)$ converges to the fixed point $p_i$.*

- *For $i \in \{1, 2\}$, there exists a constant $c_i > 0$ and a constant $K_i \in [0, 1)$ such that for any $t \geq 2$,*

$$\|x^{(t)} - p_i\|_\infty \leq K_i^t \cdot c_i \epsilon_i.$$

*Proof.* Let $w_1 := [1, 0, 0]^\top$, $w_2 := [0, 1, 0]^\top$, $w_3 := [0, 0, 1]^\top$, $b = [C, C, C]^\top$. Then it's clear that we have

$$g(\langle w_1, x \rangle + b_1) = f_1(x) = -\frac{2}{5}x_1^4 + \frac{3}{2}x_1^2$$

$$g(\langle w_2, x \rangle + b_2) = f_1(x) = -\frac{2}{5}x_2^4 + \frac{3}{2}x_2^2$$

$$g(\langle w_3, x \rangle + b_3) = f_1(x) = -\frac{2}{5}x_3^4 + \frac{3}{2}x_3^2$$

Then combing Lemma B.1, we complete the proof. $\square$

Then, through further analysis, we can construct a neural network with polynomial activation functions that has a robust fixed point as described in Theorem 4.2. Robust fixed points ensure that small perturbations or changes in the initial conditions do not lead to significant deviations in the convergence behavior of the network. We give a 3-d version of neural network.

**Lemma B.6** (A specific setting with small perturbations)**.** *Let $W := [w_1, w_2, w_3] \in \mathbb{R}^{3 \times 3}$ be a weight matrix. Let $b := [b_1, b_2, b_3] \in \mathbb{R}^3$ be a bias matrix. Let $C = \sqrt{\frac{15 + \sqrt{65}}{8}}$ be a constant. Let a polynomial activation function $g : \mathbb{R} \to \mathbb{R}$ be defined as*

$$g(z) := -\frac{2}{5}z^4 + \frac{8}{5}Cz^3 + (\frac{3}{2} - \frac{12}{5}C^2)z^2 + (\frac{8}{5}C^3 - 3C)z + 1.$$

*We define one layer of a neural network, denoted as $f$, as follows:*

$$f(x; W) := [g(\langle w_1, x \rangle + b_1), g(\langle w_2, x \rangle + b_2), g(\langle w_3, x \rangle + b_3)]^\top.$$

*Then there exists a weight matrix $W$ and a bias matrix $b$ following statements hold:*

- *The function $f(x; W)$ has at least two robust fixed point which can be found by the fixed-point iteration, and we denote them as $p_1$, $p_2$,*

- *For $i \in \{1, 2\}$, there exists $\epsilon_i \in \mathbb{R}^3$, for any initial point $x^{(0)} \in [p_{i,1} - \epsilon_{i,1}, p_{i,1} + \epsilon_{i,1}] \times [p_{i,2} - \epsilon_{i,2}, p_{i,2} + \epsilon_{i,2}] \times [p_{i,3} - \epsilon_{i,3}, p_{i,3} + \epsilon_{i,3}]$, the fixed-point iteration $x^{(t)} = f(x^{(t-1)}; W)$ converges to the robust fixed point $p_i$.*

- *For $i \in \{1, 2\}$, there exists a constant $K_i \in [0, 0.9]$ and a sufficiently large constant $m$ such that for any $t \geq 2$,*

$$\|x^{(t)} - p_i\|_\infty \leq K_i^t \cdot \|\epsilon_i\|_\infty + \frac{10}{m}$$

*Proof.* Let $m$ be a sufficiently large constant. Let $w_1 := [1, 1/m^2, 1/m^2]^\top$, $w_2 := [1/m^2, 1, 1/m^2]^\top$, $w_3 := [1/m^2, 1/m^2, 1]$, $b = [C, C, C]^\top$. Then it's clear that we have

$$g(\langle w_1, x \rangle + b_1) = -\frac{2}{5}x_1^4 + \frac{3}{2}x_1^2 + \frac{1}{m^2}h_1(x)$$

$$g(\langle w_2, x \rangle + b_1) = -\frac{2}{5}x_2^4 + \frac{3}{2}x_2^2 + \frac{1}{m^2}h_2(x)$$

$$g(\langle w_3, x \rangle + b_1) = -\frac{2}{5}x_3^4 + \frac{3}{2}x_3^2 + \frac{1}{m^2}h_3(x)$$

where for $j \in [3]$, $h_j(x)$ is a polynomial in $x_1, x_2, x_3$. It is obvious that when $m$ is sufficiently large and $\|\epsilon_i\|$ is small enough, $|\frac{1}{m^2}h_j(x)| \leq \frac{1}{m}$ for $x \in D$ and $x^{(t)} \in D$ for every $t \in \mathbb{N}$. Then we can show that for any $i \in [2], j \in [3]$,

$$|x_j^{(t)} - p_{i,j}| \leq K_{i,j}^t |x_j^{(0)} - p_{i,j}| + \frac{10}{m}$$

$$\leq K_{i,j}^t \epsilon_{i,j} + \frac{10}{m}$$

where the first comes from the result of Theorem 4.2 , the second step comes from the range of values for the initial point. Then use the definition of $\| \cdot \|_\infty$, we have

$$\|x^{(t)} - p_i\|_\infty \leq K_i^t \cdot \|\epsilon_i\|_\infty + \frac{10}{m}$$

Then we complete the proof. $\square$

## C  CASE STUDY: EXPONENTIAL ACTIVATION

In this section, we extend the discussion by constructing a neural network with exponential activation functions that do not utilize dummy variables. Similar as before, through careful design, we can ensure that the network converges to different fixed points for various initialization points.

### C.1  EXPONENTIAL ACTIVATION

In this section, we present a univariate exponential function that has at least two fixed points, which can be identified using the fixed point methods.

**Lemma C.1.** *Let $f : \mathbb{R} \to \mathbb{R}$ be a function defined as $f(x) := \exp(x^3 - 2x^2) - 1$. Then the following statements hold:*

- *The function $f$ has at least two fixed points which can be found by the fixed-point iteration, and we denoted them as $p_1, p_2$.*

- *For $i \in \{1, 2\}$, there exists a constant $\epsilon_i > 0$, for any initial point $x^{(0)} \in [p_i - \epsilon_i, p_i + \epsilon_i]$, the fixed-point iteration $x^{(t)} = f(x^{(t-1)})$ converges to the fixed point $p_i$.*

- *For $i \in \{1, 2\}$, there exists a constant $c_i > 0$ and a constant $K_i \in [0, 1)$ such that for any $t \geq 2$,*

$$|x^{(t)} - p_i| \leq K_i^t \cdot c_i \epsilon_i.$$

*Proof.* Clearly, $p_1 = 0$ and $p_2 \approx -0.9104$ are two fixed points of $f$. We show that $p_1$ and $p_2$ can be found by the fixed-point iteration and the error bounds hold.

For the fixed point $p_1 = 0$, let $\epsilon_1 = 0.1, K_1 = 0.5$, and $C_1 = 2/(1 - K_1)$. For any $x \in [p_1 - \epsilon_1, p_1 + \epsilon_1] = [-0.1, 0.1]$, we have $f(x) \in [-0.021, 0] \subseteq [-0.1, 0.1]$. Hence the first condition of Lemma A.4 is satisfied. For any $x \in [p_1 - \epsilon_1, p_1 + \epsilon_1]$, we have $|f'(x)| < K_1 = 0.5 < 1$. Hence the second condition of Lemma A.4 is satisfied. Thus by Lemma A.4, if we pick the initial point $x^{(0)} \in [p_1 - \epsilon_1, p_1 + \epsilon_1]$, we can find $p_1$ by the fixed point iteration $x^{(t)} = f(x^{(t-1)})$, and it holds that for any $t \geq 2$,

$$
\begin{aligned}
|x^{(t)} - p_1| &\leq \frac{K_1^t}{1 - K_1}|x^{(1)} - x^{(0)}| \\
&\leq \frac{K_1^t}{1 - K_1} \cdot 2\epsilon_1 \\
&= K_1^t \cdot c_1\epsilon_1,
\end{aligned}
$$

where the first step follows from Lemma A.4, the second step follows from $x^{(0)}, x^{(1)} \in [p_1 - \epsilon_1, p_1 + \epsilon_1]$, and the last step follows from $c_1 = 2/(1 - K_1)$.

For the fixed point $x_2 \approx -0.910$, let $\epsilon_2 = 0.1, K_2 = 0.85$, and $C_2 = 2/(1 - K_2)$. For any $x \in [p_2 - \epsilon_2, p_2 + \epsilon_2] = [-1.010, -0.810]$, we have $f(x) \in [-0.954, -0.842] \subseteq [-1.010, -0.810]$. Hence the first condition of Lemma A.4 is satisfied. For any $x \in [p_2 - \epsilon_2, p_2 + \epsilon_2]$, we have $|f'(x)| < K_2 = 0.85 < 1$. Hence the second condition of Lemma A.4 is satisfied. Thus by Lemma A.4, if we pick the initial point $x^{(0)} \in [p_2 - \epsilon_2, p_2 + \epsilon_2]$, we can find $p_2$ by the fixed point iteration $x^{(t)} = f(x^{(t-1)})$, and it holds that for any $t \geq 2$,

$$
\begin{aligned}
|x^{(t)} - p_2| &\leq \frac{K_2^t}{1 - K_2}|x^{(1)} - x^{(0)}| \\
&\leq \frac{K_2^t}{1 - K_2} \cdot 2\epsilon_2 \\
&= K_2^t \cdot c_2\epsilon_2,
\end{aligned}
$$

where the first step follows from Lemma A.4, the second step follows from $x^{(0)}, x^{(1)} \in [p_2 - \epsilon_2, p_2 + \epsilon_2]$, and the last step follows from $c_2 = 2/(1 - K_2)$.

Therefore, we complete the proof. $\square$

Next, we prove two lemmas which are useful to construct a neural network with a exponential activation function.

**Lemma C.2.** *Let*

$$
f_1(x) := \exp(\langle u, x\rangle^3 + (-2 - 3C)\langle u, x\rangle^2 + (3C^2 + 4C)\langle u, x\rangle + \ln 2) - 1
$$

*where $C \approx -2.15$, $u := [1, 1, C - 1]$. Let $f_2(x) := 1, f_3(x) := 1$. Then there exists a function $g(z) := \exp(a_3 z^3 + a_2 z^2 + a_1 z + a_0) - 1$, where $a_0, a_1, a_2, a_3 \in \mathbb{R}$ such that for some $w_1, w_2, w_3 \in \mathbb{R}^3$, we have $g(\langle w_1, x\rangle) = f_1(x)$, $g(\langle w_2, x\rangle) = f_2(x)$ and $g(\langle w_3, x\rangle) = f_3(x)$. Moreover we have*

$$
g(z) := \exp(z^3 + (-2 - 3C)z^2 + (3C^2 + 4C)z + \ln 2) - 1.
$$

*and $w_1 = [1, 1, C - 1]^\top, w_2 = [0, 0, 0]^\top, w_3 = [0, 0, 0]^\top$.*

*Proof.* Let $g(z) := \exp(z^3 + (-2 - 3C)z^2 + (3C^2 + 4C)z + \ln 2) - 1$. Let $w_1 := [1, 1, 1 - C]^\top, w_2 := [0, 0, 0]^\top, w_3 := [0, 0, 0]^\top$. Then it is clear that we have

$$
\begin{aligned}
g(\langle w_1, x\rangle) &= f_1(x), \\
g(\langle w_2, x\rangle) &= f_2(x), \\
g(\langle w_3, x\rangle) &= f_3(x).
\end{aligned}
$$

Thus we complete the proof. $\square$

**Lemma C.3.** *Let*

$$f_1(x) := \exp(\langle u, x\rangle^3 + (-2 - 3C)\langle u, x\rangle^2 + (3C^2 + 4C)\langle u, x\rangle + \ln 2) - 1$$

*where $C \approx -2.15$, $u := [1, 1, C - 1]$. It holds that*

$$f_1([x_1, 1, 1]^\top) = \exp(x_1^3 - 2x_1^2) - 1.$$

*Proof.* Note that $\langle u, [x_1, 1, 1]^\top\rangle = x_1 + C$. Hence

$$f_1([x_1, 1, 1]^\top) = \exp((x_1 + C)^3 + (-2 - 3C)(x_1 + C)^2 + (3C^2 + 4C)(x_1 + C) + \ln 2) - 1$$
$$= \exp(x_1^3 - 2x_1^2) - 1$$

where the first step follows from $\langle u, [x_1, 1, 1]^\top\rangle = x_1 + C$, and the second step follows from expanding the higher-order terms and simplifying them. $\qquad\square$

## C.2 EXTEND TO NEURAL NETWORKS

We are now prepared to construct a neural network with exponential activation functions. By carefully designing the weight matrix, we can ensure that the network converges to different fixed points for different initialization. We give a 3-d neural network version.

**Lemma C.4.** *Let $W := [w_1, w_2, w_3] \in \mathbb{R}^{3\times 3}$ be a weight matrix. Let $C := -2.15$ be a constant. Let a exponential activation function $g : \mathbb{R} \to \mathbb{R}$ be defined as*

$$g(z) := \exp(z^3 + (-2 - 3C)z^2 + (3C^2 + 4C)z + \ln 2) - 1.$$

*We define one layer of a neural network, denoted as $f$, as follows:*

$$f(x; W) := [g(\langle w_1, x\rangle), g(\langle w_2, x\rangle), g(\langle w_3, x\rangle)]^\top.$$

*Then there exists a weight matrix $W$ following statements hold:*

- *The function $f(x; W)$ has at least two fixed points which can be found by the fixed-point iteration, and we denoted them as $p_1, p_2$.*

- *For $i \in \{1, 2\}$, there exists a constant $\epsilon_i > 0$, for any initial point $x^{(0)} \in [p_{i,1} - \epsilon_i, p_{i,1} + \epsilon_i] \times \{1\} \times \{1\}$, the fixed-point iteration $x^{(t)} = f(x^{(t-1)}; W)$ converges to the fixed point $p_i$.*

- *For $i \in \{1, 2\}$, there exists a constant $c_i > 0$ and a constant $K_i \in [0, 1)$ such that for any $t \geq 2$,*

$$\|x^{(t)} - p_i\|_\infty \leq K_i^t \cdot c_i\epsilon_i.$$

*Proof.* The lemma holds by combining Lemma C.1, Lemma C.2, and Lemma C.3. $\qquad\square$

## C.3 EXTENSION TO $d$-DIMENSIONAL CASE

Next, we extend Lemma C.2 to $d$-dimensional case.

**Lemma C.5.** *Let*

$$f_1(x) := \exp(\langle u, x\rangle^3 + (-2 - 3C)\langle u, x\rangle^2 + (3C^2 + 4C)\langle u, x\rangle + \ln 2) - 1$$

*where $C \approx -2.15$, $u := [1, \frac{1}{d-1}, ..., \frac{1}{d-1}, \frac{1}{d-1} + C - 1]^\top \in \mathbb{R}^d$. For all $i \in \{2, ..., d\}$, let $f_i(x) := 1$. Then there exists a function $g(z) := \exp(a_3 z^3 + a_2 z^2 + a_1 z + a_0) - 1$, where $a_0, a_1, a_2, a_3 \in \mathbb{R}$ such that for some $w_1, w_2, ..., w_d \in \mathbb{R}^d$, we have that for all $i \in [d]$*

$$g(\langle w_i, x\rangle) = f_i(x)$$

*Moreover we have*

$$g(z) = \exp(z^3 + (-2 - 3C)z^2 + (3C^2 + 4C)z + \ln 2) - 1$$

*and $w_1 = [1, \frac{1}{d-1}, ..., \frac{1}{d-1}, \frac{1}{d-1} + C - 1] \in \mathbb{R}^d$, for all $i \in \{2, ..., d\}$, $w_i = [0, ..., 0] \in \mathbb{R}^d$.*

*Proof.* Let $g(z) = \exp(z^3 + (-2 - 3C)z^2 + (3C^2 + 4C)z + \ln 2) - 1$. Let $w_1 := [1, \frac{1}{d-1}, ..., \frac{1}{d-1}, \frac{1}{d-1} + C - 1] \in \mathbb{R}^d$. For all $i \in \{2, ..., d\}$, let $w_i = [0, ..., 0] \in \mathbb{R}^d$. Then it is clear that for all $i \in [d]$ we have

$$g(\langle w_i, x \rangle) = f_i(x)$$

Thus we complete the proof. $\qquad \square$

Then, we extend Lemma C.3 to $d$-dimensional case.

**Lemma C.6.** *Let*

$$f_1(x) := \exp(\langle u, x \rangle^3 + (-2 - 3C)\langle u, x \rangle^2 + (3C^2 + 4C)\langle u, x \rangle + \ln 2) - 1$$

*where* $C \approx -2.15$, $u := [1, \frac{1}{d-1}, ..., \frac{1}{d-1}, \frac{1}{d-1} + C - 1]^\top \in \mathbb{R}^d$. *It holds that*

$$f_1([x_1, \mathbf{1}_{d-1}^\top]^\top) = \exp(x_1^3 - 2x_1^2) - 1$$

*Proof.* Note that $\langle u, [x_1, \mathbf{1}_{d-1}^\top]^\top \rangle = x_1 + C$. Hence

$$f_1([x_1, \mathbf{1}_{d-1}^\top]^\top) = \exp((x_1 + C)^3 + (-2 - 3C)(x_1 + C)^2 + (3C^2 + 4C)(x_1 + C) + \ln 2) - 1$$
$$= \exp(x_1^3 - 2x_1^2) - 1$$

where the first step follows from $\langle u, [x_1, \mathbf{1}_{d-1}^\top]^\top \rangle = x_1 + C$, and the second step follows from expanding the higher-order terms and simplifying it. $\qquad \square$

Then, we extend Lemma C.4 to $d$-dimensional case.

**Lemma C.7.** *Let* $W := [w_1, w_2, ..., w_d] \in \mathbb{R}^{d \times d}$ *be a weight matrix. Let* $C := -2.15$ *be a constant. Let a exponential activation function* $g : \mathbb{R} \to \mathbb{R}$ *be defined as*

$$g(z) := \exp(z^3 + (-2 - 3C)z^2 + (3C^2 + 4C)z + \ln 2) - 1.$$

*We define one layer of a neural network, denoted as* $f$, *as follows:*

$$f(x; W) := [g(\langle w_1, x \rangle), g(\langle w_2, x \rangle), ..., g(\langle w_d, x \rangle)]^\top.$$

*Then there exists a weight matrix* $W$ *following statements hold:*

- *The function* $f(x; W)$ *has at least two fixed points which can be found by the fixed-point iteration, and we denoted them as* $p_1, p_2$.

- *For* $i \in \{1, 2\}$, *there exists a constant* $\epsilon_i > 0$, *for any initial point* $x^{(0)} \in [p_{i,1} - \epsilon_i, p_{i,1} + \epsilon_i] \times \{1\} \times \{1\}$, *the fixed-point iteration* $x^{(t)} = f(x^{(t-1)}; W)$ *converges to the fixed point* $p_i$.

- *For* $i \in \{1, 2\}$, *there exists a constant* $c_i > 0$ *and a constant* $K_i \in [0, 1)$ *such that for any* $t \geq 2$,

$$\|x^{(t)} - p_i\|_\infty \leq K_i^t \cdot c_i \epsilon_i.$$

*Proof.* The lemma holds by combining Lemma C.1, Lemma C.5, and Lemma C.6. $\qquad \square$

## LLM USAGE DISCLOSURE

LLMs were used only to polish language, such as grammar and wording. These models did not contribute to idea creation or writing, and the authors take full responsibility for this paper's content.

