# OpenReview forum: "Advancing the understanding of fixed point iterations in deep neural networks: a detailed analytical study"
_ICLR.cc/2026/Conference — Submitted to ICLR 2026_

### Official Review · Reviewer_53pi · 2025-10-24

**Soundness:** 3
**Presentation:** 2
**Contribution:** 1
**Rating:** 2
**Confidence:** 4

**Summary:**

This paper analyzes the fixed points of "looped neural networks," which are feed-forward networks where weights are shared across layers.  Looped neural networks can be viewed as a type of recurrent neural network whose hidden state at t=0 is initialized to the input data, and whose final hidden state at t=T is treated as the output prediction, without any separate connections for sequential input/output. This paper is motivated by recent looped neural network architectures where the shared layer is a transformer block, and various useful computations are performed by iterating the block until convergence to a fixed point.

The paper focuses on a simpler and more classical looped neural network architecture, where the shared layer is a linear transformation with Lipschitz constant less than 1, followed by an element-wise non-linear activation function.  The paper proves that two specific instances of this architecture - one whose activation function is a polynomial, another whose activation is a polynomial composed with an exponential - can have a number of fixed points that is exponential in the size of the hidden layer.  They also prove that convergence to a fixed point is relatively stable under small perturbations of the activation vector at each time-step.  Some small-scale simulation experiments corroborate the theoretical results.

**Strengths:**

- This paper focuses on an interesting, (re-)emerging area of recurrent neural networks (in this case, looped transformers) and analysis of their fixed points.  Improving our theoretical understanding of RNN fixed points may be helpful in their mechanistic interpretation and design.

- The proofs appear to be sound and are supported empirically, which is a plus.

- Examining the case of convergence under noise is interesting and relevant to adversarial robustness.

- Complete proofs and experimental code are included in appendix and supplementary material, which promotes reproducibility.

**Weaknesses:**

- Experimental results are very limited - just a single paragraph and some figures - and only applies to synthetic data.  Including real-world tasks and data would be more compelling and relevant to the looped neural network research community.

- I question the significance of the theoretical results, for the following reasons:
    - The results only show that many fixed points exist; it is unclear exactly if/how this would inform neural network design.
    - The results only apply to a classical type of layer, namely linear composed with element-wise activation function.  Fixed points of this layer have already been well-studied (e.g. [1,2,3,4] below), and they are not relevant to the looped transformer networks that are cited as motivation for this paper.
    - The activation functions considered (exponential of/polynomial) are not common in practice. In fact, similar results hold for a more common activation function (hyperbolic tanh) when the weight matrix is an identity matrix scaled by a number larger than 1; such an example might make for a more relevant case study.
    - Theorem 4.1 seems to be a direct application of Banach's fixed point theorem, limiting its novelty.

- There is some imprecise word choice that made it difficult for me to understand exactly what was being claimed:
    - In 3.2, "we first introduce the fixed point iteration problem" is followed by the definition of a fixed point, not the definition of an iterative process
    - The "noise term" h(x) in Theorem 4.2 appears to be a deterministic function of x.
    - Theorem 5.1 describes W and b as "noisy parameters" but they do not appear to be random variables.

[1] Katz GE, Reggia JA. Using directional fibers to locate fixed points of recurrent neural networks. IEEE transactions on neural networks and learning systems. 2017 Aug 24;29(8):3636-46.

[2] Eisenmann L, Monfared Z, Göring N, Durstewitz D. Bifurcations and loss jumps in RNN training. Advances in Neural Information Processing Systems. 2023 Dec 15;36:70511-47.

[3] Sussillo D, Barak O. Opening the black box: low-dimensional dynamics in high-dimensional recurrent neural networks. Neural computation. 2013 Mar 1;25(3):626-49.

[4] Golub MD, Sussillo D. FixedPointFinder: A Tensorflow toolbox for identifying and characterizing fixed points in recurrent neural networks. Journal of open source software. 2018 Nov 1;3(31):1003.

**Questions:**

- Some theorems involve unexplained constants, such as 0.95 and 20 in Theorem 4.2 (likewise in Theorem 5.1).  Are these constants essential, or would similar results hold for other values of these constants?
- Theorem 5.1 says K_i in [0,.9) but layer states K_j = 0.92.  Is this a contradiction, or do these variables denote different quantities?

---

### Official Review · Reviewer_cLiS · 2025-10-25

**Soundness:** 2
**Presentation:** 2
**Contribution:** 2
**Rating:** 2
**Confidence:** 3

**Summary:**

The submitted work studies fixed points in a special type of neural networks called looped networks. Looped networks are neural networks but with the same weights in every layer. The motivation here is to understand how repeated application of a layer can result in a fixed point: $f(x) = x$ (simplified for intuition). The authors provide conditions under which looped networks exhibit fixed points (the activations do not change in subsequent layers). They then extend their analysis to the case of noise perturbations at each layer and other specific cases.

**Strengths:**

* The authors introduce a novel mathematical analysis for analyzing fixed points in Looped networks
* The paper is clearly written and provides intuitions behind the main results using simple examples (Figure 1 and 2)

**Weaknesses:**

**Practical implications**: While the authors give examples of relevance of fixed point iterations, such as
* Adjacent layers may perform identical operations
* Skipping layers during inference,

they are not directly related to the fixed point analysis performed in this work. I would request the authors to help me understand the connection better.

**No Real Network Experiments**: The experiments are limited to toy synthetic setups and its unclear if they have any real-world implications.

**Questions:**

* Can you comment on how strong is the derivative assumption in Theorem 4.1? I am unsure if its expected to holds generally.
* Do trained neural networks exhibit fixed points? Can we take individual layers of a LLM and study the fixed points? I believe the authors can study the recurrent blocks of recurrent Transformers [2]
* Is there a way to use the fixed point analysis to skip layers during inference?
* Can the authors suggest improvements in existing networks to avoid fixed point collapse?


[1] Exponential expressivity in deep neural networks through transient chaos, 2016

[2] Scaling up test-time compute with latent reasoning: A recurrent depth approach, 2025

---

### Official Review · Reviewer_3H88 · 2025-10-30

**Soundness:** 3
**Presentation:** 3
**Contribution:** 3
**Rating:** 6
**Confidence:** 2

**Summary:**

The paper studies fixed-point iterations in looped neural networks, establishing a sufficient condition for the existence of multiple fixed points (Theorem 4.1) and their variant under noise perturbations (Theorem 4.2). The authors also present illustrative case studies using polynomial and exponential activation functions, and simple simulations validating their theory.

**Strengths:**

The paper is clearly written and well structured. It offers a solid theoretical contribution by generalizing Banach fixed-point analysis to multiple fixed points and robustness under noise. The related work section provides a coherent overview and motivation for studying fixed points in neural networks.

**Weaknesses:**

The empirical section relies on toy examples with artificial activation functions. It would be much stronger to include experiments with more practical architectures or common nonlinearities (e.g., ReLU) to show broader applicability. The discussion of the potential implications for real-world or large-scale networks remains somewhat abstract.

**Questions:**

I can see from the paper that fixed-point analysis is an emerging subfield, but being non-expert in this subfield, I personally feel it is still a niche direction in deep learning theory. Could the authors clarify what concrete algorithmic benefits or new analytical tools their framework could bring for mainstream architectures (e.g., recurrent, equilibrium, or transformer models)? More discussion on why multiple fixed points matter for understanding or improving deep models would make the paper’s impact clearer to a broader audience.

---

### Official Review · Reviewer_DZqY · 2025-10-31

**Soundness:** 3
**Presentation:** 3
**Contribution:** 2
**Rating:** 6
**Confidence:** 3

**Summary:**

This paper provides a theoretical analysis of fixed-point iterations in looped neural networks (LNNs) — architectures where the same layer (with shared weights) is repeatedly applied. The study formalizes conditions for the existence and robustness of multiple fixed points and relates these to practical phenomena observed in deep or residual-like networks.

Core contributions:

General Theorem (Theorem 4.1):
Establishes sufficient conditions for multiple fixed points in LNNs based on Banach’s fixed-point theorem, via contractivity within disjoint input domains D_i.

Robust Fixed-Point Theorem (Theorem 4.2):
Extends Banach’s theorem to include perturbations/noise, modeling the effect of residual connections. Provides quantitative error bounds showing that iteration remains convergent if |f (x) | < 0.95 even with additive noise up to 1/m.

The paper links looped neural dynamics to fixed-point iteration theory, establishing mathematical foundations for phenomena observed in Deep Equilibrium Models (DEQ) and looped Transformers.

**Strengths:**

Clear mathematical framing:
The use of Banach’s contraction principle and its noisy variant gives a solid analytical grounding for convergence of LNNs.
Proofs are rigorous and fully detailed in Appendix A–C.

Novel sufficient conditions:
Theorem 4.1’s condition using disjoint contractive regions is an elegant way to guarantee multiple attractors in high-dimensional networks—extending traditional single-fixed-point analyses.

Robustness analysis (Theorem 4.2):
Modelling small perturbations as noise provides a bridge between theoretical analysis and residual or perturbed networks, which is practically meaningful.

Comprehensive case studies:
Demonstrating 2^d fixed points under polynomial/exponential activations is conceptually neat and mathematically clean. Figure 3’s experiment (p. 8) shows convergence of 1024 trajectories to distinct attractors, visually confirming the results.

**Weaknesses:**

Restrictive examples:
The polynomial and exponential activations are synthetic, designed specifically to satisfy the theorems. It remains unclear whether common activations (ReLU, tanh, GELU) exhibit similar multi-fixed-point structures.

Generality of disjoint-region assumption:
Theorem 4.1 requires explicit partitioning of the input space into disjoint D_i where contractivity holds. For most realistic networks, such partitioning with constants K_i < 1 are hard to verify or may fail.

Exponential-growth interpretation:
While the existence of 2^d fixed points is mathematically correct for specially chosen g (x) in practical networks such exponentially many stable equilibria may not exist; the relevance to more realistic networks is not discussed.

**Questions:**

1) Can you provide examples using standard activations (ReLU, tanh) where contractivity and multiple fixed points approximately hold?

2) How do your sufficient conditions relate to the Jacobian spectral radius used in DEQ stability analyses?

3) The theorems are proved for constant weights. Can you comment on how these results could be lifted to weights if they were updated across iterations?

4) Does the 2^d fixed-point structure improves performance or learning efficiency in LNNs?

5) How could these results be lifted to DEQ analysis in Looped Transformers as suggested in the tex?

The reason why I did not put a higher score is because it's not clear to me what is the impact of these results so if you could provide arguments towards this, I would be willing to increase my score.

---

### Meta-Review · Area_Chair_XmAJ · 2026-01-06

**Summary:**

This paper studies fixed-point iteration behavior in looped neural networks (shared-weight layers), aiming to put recent empirical observations about “hidden states stabilizing” on a more formal footing. The main technical content is a sufficient-condition theorem for multiple fixed points via contractive behavior on disjoint regions, plus a robustness variant under small perturbations/noise. Two reviewers find the theory clean and reasonably presented (borderline accept), while two reviewers are negative mainly due to limited novelty (largely a Banach-style application), highly synthetic case studies/activations, and a lack of convincing relevance to practical architectures like looped transformers or DEQ beyond high-level motivation.

**Reviewer Concerns:**

Concerns addressed by the rebuttal:
I did not see a substantive rebuttal in the available forum snapshot, so I cannot credit specific rebuttal resolutions.

Concerns still outstanding:
1) Practical relevance remains unclear. Reviewers repeatedly ask for either real-network evidence (or at least more realistic nonlinearities) and a clearer story for what the “many fixed points” result implies for design/understanding of modern looped models.
2) Novelty/impact is the sticking point. The main theorems are seen as natural extensions/applications of classical contraction mapping tools, with the most striking examples relying on carefully constructed polynomial/exponential activations.
3) Empirical support is very limited and purely synthetic, which makes it hard to judge whether the proposed analytical lens will matter for the looped-transformer/DEQ community the paper cites as motivation.

**Reviewer Scores:**

1) Reviewer DZqY (score 6): likely unchanged at 6. They already indicate they might increase if impact is clearer, but without visible rebuttal and with experiments still synthetic, a change is unlikely.
2) Reviewer 3H88 (score 6): likely unchanged at 6 for similar reasons; they value the clean theory but still describe the impact discussion as abstract.
3) Reviewer cLiS (score 2): likely unchanged at 2. Their main objections are about missing real-network implications/experiments and unclear connection to practical claims.
4) Reviewer 53pi (score 2): likely unchanged at 2. Their critique centers on significance/novelty and mismatch to the looped-transformer motivation, which would require substantial new evidence or reframing to change.

---

### Decision · Program_Chairs · 2026-01-26

Reject